# Differentially Expressed Genes in Rat Brain Regions with Different Degrees of Ischemic Damage

**DOI:** 10.3390/ijms26052347

**Published:** 2025-03-06

**Authors:** Ivan B. Filippenkov, Yana Yu. Shpetko, Vasily V. Stavchansky, Alina E. Denisova, Vadim V. Yuzhakov, Natalia K. Fomina, Leonid V. Gubsky, Svetlana A. Limborska, Lyudmila V. Dergunova

**Affiliations:** 1Laboratory of Human Molecular Genetics, National Research Centre “Kurchatov Institute”, Kurchatov Sq. 2, Moscow 123182, Russia; yana.sch2014@yandex.ru (Y.Y.S.); bacbac@yandex.ru (V.V.S.); limbor.img@yandex.ru (S.A.L.); dergunova-lv.img@yandex.ru (L.V.D.); 2Department of Neurology, Neurosurgery and Medical Genetics, Pirogov Russian National Research Medical University, Ostrovitianov Str. 1, Moscow 117997, Russia; dalina543@gmail.com (A.E.D.); gubskii@mail.ru (L.V.G.); 3A. Tsyb Medical Radiological Research Center—Branch of the National Medical Research Radiological Center of the Ministry of Health of the Russian Federation, Koroleva Str. 4B, Obninsk 249036, Russiankfomina@rambler.ru (N.K.F.); 4Federal Center for the Brain and Neurotechnologies, Federal Biomedical Agency, Ostrovitianov Str. 1, Building 10, Moscow 117997, Russia

**Keywords:** ischemic stroke, brain damage, ipsilateral striatum, frontal cortex, RNA-seq, tMCAO, gene expression, gene network

## Abstract

Ischemic stroke is a multifactorial disease that leads to brain tissue damage and severe neurological deficit. Transient middle cerebral artery occlusion (tMCAO) models are actively used for the molecular, genetic study of stroke. Previously, using high-throughput RNA sequencing (RNA-Seq), we revealed 3774 differentially expressed genes (DEGs) in the penumbra-associated region of the frontal cortex (FC) of rats 24 h after applying the tMCAO model. Here, we studied the gene expression pattern in the striatum that contained an ischemic focus. Striatum samples were obtained from the same rats from which we previously obtained FC samples. Therefore, we compared DEG profiles between two rat brain tissues 24 h after tMCAO. Tissues were selected based on magnetic resonance imaging (MRI) and histological examination (HE) data. As a result, 4409 DEGs were identified 24 h after tMCAO in striatum. Among them, 2609 DEGs were overlapped in the striatum and FC, whereas more than one thousand DEGs were specific for each studied tissue. Furthermore, 54 DEGs exhibited opposite changes at the mRNA level in the two brain tissues after tMCAO. Thus, the spatial regulation of the ischemic process in the ipsilateral hemisphere of rat brain at the transcriptome level was revealed. We believe that the targeted adjustment of the genome responses identified can be the key for the induction of regeneration processes in brain cells after stroke.

## 1. Introduction

Ischemic stroke is a multifactorial disease that leads to brain tissue damage and severe neurological deficit. Tens of millions of people suffer from stroke every year [1,2]. Stroke remains the second leading cause of death and the third leading cause of death and disability-adjusted life years (DALYs) in the world. From 1990 to 2021, the burden increased substantially (70.0% increase in stroke incidence, 44.0% increase in deaths from stroke, 86.0% increase in stroke prevalence, and 32% increase in DALYs) [3]. The incidence, DALY rates, and mortality of stroke were 141.553, 1886.196, and 87.454 per 100,000 persons in 2021 [4]. In addition, ischemic stroke may be a risk factor for Alzheimer’s disease [5,6]. For example, both Alzheimer’s disease and cerebral ischemia lead to the accumulation of beta-amyloid peptides in the extracellular space of the brain [7,8]. In this context, the regulation of ionic strength, protein concentration [9], and certain divalent cations [10] may be important in brain cells. Cerebral ischemia causes a cascade of biochemical changes in brain tissues [11]. The induction of glutamate-mediated pathways triggers cell death by massive calcium influx following vessel occlusion [12,13], and a lack of oxygen causes failure in the respiratory chain and mitochondrial function [14,15]. These changes lead to an inflammatory reaction [16]. The violation of the function of the blood–brain barrier allows the invasion of immune cells and pathogenic factors into brain cells [17,18]. Different studies have succeeded in identifying the modulation of the expression pattern of many genes involved in metabolic cell activity, neurotransmission, inflammation, the immune response, apoptosis, and the stress response in the brain regions of animal models of stroke. The rodent ischemia model systems include variants of embolic occlusion, photothrombosis, endothelin-1-induced vasoconstriction; however, permanent middle cerebral artery occlusion (pMCAO) and transient middle cerebral artery occlusion (tMCAO) are more often used for stroke studies [19,20,21,22,23,24]. The pMCAO model simulates most clinical stroke cases that are not treated promptly, leaving the blocked artery obstructed. At the same time, an important component of tMCAO is reperfusion, developing against the background of blood flow restoration. This model mainly reflects events after anti-stroke treatment with thrombolytic drugs [25]. Many studies have provided a molecular basis for understanding the mechanisms of neuroprotective stroke therapy, which is currently being actively developed [23,26,27,28,29,30,31].

Currently, studies in the field of the spatial and temporal regulation of ischemic processes in brain zones with different degrees of ischemic injury are actively developing [22,32,33,34]. The ischemic core is the most affected area, while the penumbra retains some metabolic and polarization capacity [35]. Following ischemic injury, the structure and function of brain cells modified include the formation of glial scars [36]. The pathological environment and recruiting microglia influence nearby astrocytes and oligodendrocytes, causing transcriptional, morphological, and metabolic changes in cells [37]. A question arises about the structure of the affected and surrounding brain regions and the genomic and metabolic signatures of the cells included in them.

The genomic component occupies a significant place in the pathogenesis, diagnosis, and treatment of ischemic stroke. The high-throughput RNA sequencing (RNA-Seq) method provides measurements of levels of transcripts and their isoforms in biological samples on a genome-wide scale [38]. Thus, RNA-Seq makes it possible to identify genes that show significant differences in expression levels between two or more groups [39,40]. These genes are called differentially expressed genes (DEGs). Previously, using RNA-Seq, we identified thousands DEGs with cut-offs more than 1.5 and *Padj* < 0.05 in the penumbra-associated region of the frontal cortex (FC) under IR 24 h after tMCAO with 90 min of occlusion [31]. This tMCAO model reflects ischemia–reperfusion (IR) processes in humans after strokes, including thrombolytic therapy [30,41]. In the present study, using RNA-Seq, we studied spatial gene expression patterns in rat brain 24 h after tMCAO and compared profiles of DEGs of the FC and striatum that contained an ischemic focus. Tissues were selected based on magnetic resonance imaging (MRI) and histological examination (HE) data and obtained from the same rats. The localization of the region of FC and striatum is shown in Appendix A. As a result, 2609 DEGs overlapped in striatum and FC, whereas 1800 and 1165 DEGs were specific to the striatum and FC, respectively. Furthermore, 54 DEGs exhibited opposite changes at the mRNA level in the two brain tissues after tMCAO. Functional analysis revealed genome-wide associations of DEGs with inflammatory, neurosignaling, and metabolic systems in accordance with studied tissues. Thus, the spatial regulation of the ischemic process in the ipsilateral hemisphere of rat brain at the transcriptome level was revealed after tMCAO in rats. We believe that the targeted adjustment of genome responses identified can be the key for the induction of regeneration processes in brain cells after stroke and following its treatment.

## 2. Results

### 2.1. Histological Examination of Brain Samples

Histological examination (HE) with Nissl staining of the ipsilateral striatum is shown in Figure 1a,b. Rats in IR groups had ischemic injury areas with ipsilateral hemispheric (subcortex plus cortex) localization in the brain of rats 24 h after tMCAO (Figure 1a). The striatum contained necrotic and partially penumbra zones (Figure 1b).

### 2.2. RNA-Seq Analysis of the Effect of IR on the Striatum of Rats 24 h After tMCAO

Using RNA-Seq, we analyzed the transcriptional activity of the mRNAs for 17,367 genes in the striatum of rats 24 h after tMCAO. In a pairwise comparison of the RNA-Seq data for IR-s vs. SO-s (Figure 1c, Appendix A), we identified significant changes at the mRNA level for 4409 DEGs with 2364 up- and 2045 downregulated genes. The volcano plot in Figure 1d demonstrates the variations in mRNA expression levels between the IR-s and SO-s comparison groups. Note that the top five most upregulated genes in response to IR were increased by ≥109 times including *Ccl2*, *Socs3*, *Ptx3*, *Hspa1*, and *Lgals3*. At the same time, the top five downregulated DEGs were *Btbd17*, *Hes5*, *Neurod2*, *Plk5*, and *Ercc2*.

RT–PCR analysis of the expression of four genes (*Mmp9*, *Nos3*, *il4r*, and *Vegfa*) was used to verify the RNA-Seq results for IR-s vs. SO-s. The characterization of the primers is shown in Appendix A. The real-time RT–PCR results confirmed the RNA-Seq data (Appendix A).

### 2.3. Comparison of RNA-Seq Results in Different Groups in Striatum and FC 24 h After tMCAO

Previously, using RNA-Seq, we identified thousands DEGs with cut-offs of more than 1.5 and *Padj* < 0.05 in the FC under IR 24 h after tMCAO [31]. Here, our analysis found that IR action modulated both overlapped and unique gene expression profiles in the striatum and FC. So, under IR, we identified 2609 overlapping DEGs in the FC (IR-f vs. SO-f) and ipsilateral striatum (IR-s vs. SO-s) (Figure 2a, Appendix A). Venn diagrams with only upregulated genes and only downregulated genes under IR in both tissues are presented in Figure 2b,c, respectively. Predominantly, overlapping genes co-directly changed their mRNA level in both tissues 24 h after tMCAO. Namely, among such genes, we found 1334 up- and 1221 downregulated DEGs in both tissues. Thus, 2555 out of 2609 DEGs were co-directly changed in the FC and striatum 24 h after tMCAO. The top 10 overlapping genes that had the greatest fold change in IR-s vs. SO-s included *Ccl2*, *Socs3*, *Hspa1a*, *Lgals3*, and *Spp1* as up- and *Ddc*, *Btbd17*, *Hes5*, *Neurod2*, and *Plk5* as downregulated DEGs in both tissues.

In addition, the Venn diagram in Figure 2a presents two relative complements for IR-s vs. SO-s and IR-f vs. SO-f pairwise comparisons, respectively. So, we found 1800 DEGs that altered the expression levels only in the striatum that contained an infarction focus. The top 10 genes from this list with the greatest fold change in IR-s vs. SO-s included *Itgad*, *Inmt*, *Clec7a*, *Runx3*, and *Il6* as up- and *Krt85*, *Ctf2*, *Asb18*, *Aanat*, and *Atp1a4* as downregulated genes in IR-s vs. SO-s (Figure 2e). The top 10 out of 1165 genes from the IR-f vs. SO-f relative complements included *Dspp*, *Rxfp2*, *Slc6a5*, *Ccl11*, and *Dkk2* as up- and *Dact2*, *Gypc*, *Vwa5b1*, *Gzmm*, and *Lrrc17* as downregulated genes in IA-f vs. IR-f (Figure 2f). A full list of DEGs unique to the IR-s vs. SO-s and IR-f vs. SO-f pairwise comparisons are shown in Appendix A, respectively.

The hierarchical cluster analysis of all DEGs in the IR-s vs. SO-s and IR-f vs. SO-f is illustrated in Figure 2g. The common features of the differential expression profiles of these comparison groups reflect the effects of IR in both rat brain tissues. At the same time, individual differences between the groups seem to characterize the specific genome responses in the FC and striatum at the transcriptome level.

### 2.4. DEGs Exhibited Opposite Changes at the mRNA Level in the Two Brain Tissues 24 h After tMCAO

Among the overlapping genes between IR-s and SO-s and IR-f and SO-f were those that showed opposite changes in expression in the striatum and FC. Namely, we found 19 DEGs that were upregulated in the IR-s vs. SO-s and downregulated in the IR-f vs. SO-f pairwise comparisons (Figure 3a). Additionally, there were 35 DEGs that were downregulated in the IR-s vs. SO-s and upregulated in the IR-f vs. SO-f pairwise comparisons (Figure 3b). Thus, in total, 54 DEGs exhibited opposite changes at the mRNA level in the two brain tissues 24 h after tMCAO. Figure 3c shows the top 10 of these genes identified with the greatest fold change in IR-s vs. SO-s. Among them are the *Rsph10b*, *Rsph1*, *Ehhadh*, *Bace2*, and *Pgap1* genes that belong to the first group, as well as the *Igsf21*, *Stk32a*, *Kremen1*, *Cnr1*, and *Htr1d* genes that belong to the second group. Additionally, Appendix A illustrates that 54 DEGs exhibited opposite changes at the mRNA level in the two brain tissues 24 h after tMCAO, observed using hierarchical cluster analysis.

### 2.5. Functional Annotations of DEGs Altered in Different Comparison Groups

Using DAVID v.2021, a Kyoto Encyclopedia of Genes and Genomes (KEGG) pathway-enrichment analysis of DEGs in the IR-s vs. SO-s pairwise comparison was carried out. As a result, 151 KEGG pathways were identified with *Padj* < 0.05 for DEGs in IR-s vs. SO-s (Figure 4a, Appendix A). Previously, a similar functional analysis was performed for DEGs in the FC in IR-f vs. SO-f, where 134 KEGG pathways were identified [31]. Figure 4a shows a comparison of pathway sets between the IR-s and SO-s and IR-f and SO-f pairwise comparisons using Venn diagram. As a result, 120 pathways overlapped between the IR-s and SO-s and IR-f and SO-f groups. We assigned these pathways to the first cluster (Pathway Cluster 1, PC1). The top five overlapping pathways with the most significant *Padj* value in IR-s vs. SO-s included the glutamatergic synapse, lipid and atherosclerosis, and mitogen-activated protein kinase (MAPK) signaling pathways and others (Figure 4b). Genes associated with these pathways predominantly showed similar differential expression profiles in both the FC and striatum.

It should be noted that 31 and 14 pathways lie in relative complements for IR-s vs. SO-s and IR-f vs. SO-f, respectively, in the Venn diagram (Figure 4a). We assigned the pathways of the IR-s vs. SO-s-relative complement to the second cluster (Pathway Cluster 2, PC2). The top five out of those in PC2 included the sphingolipid signaling pathway, gonadotropin-releasing hormone (GnRH) secretion, protein processing in the endoplasmic reticulum, and others (Figure 4c). Additionally, we revealed cell cycle, antigen processing and presentation, DNA replication, phagosomes, and other functional annotations. Listed pathways were specific to DEG sets in the striatum. Meanwhile, the list of the top five specific pathways in the FC included neuroactive ligand–receptor interaction, the biosynthesis of cofactors, cysteine and methionine metabolism, and other signaling pathways (Figure 4d). We assigned such pathways to the third cluster (Pathway Cluster 3, PC3).

Then, all pathways were additionally classified according to the differential gene expression profiles in the IR-s vs. SO-s and IR-f vs. SO-f pairwise comparisons using Heatmapper (Appendix A). The classification was based on the difference between upregulated and downregulated DEGs in each comparison group. The most significant increase in gene expression in both tissues was observed in the group of pathways associated with the immune system, as well as ribosome biogenesis, apoptosis, the vascular endothelial growth factor (VEGF) signaling pathway, and cellular senescence. Concomitantly, the most significant decrease in gene expression was in the group of pathways associated with neurotransmission.

Additionally, using the Database for Annotation, Visualization, and Integrated Discovery (DAVID) functional annotation tool, we analyzed 54 DEGs that exhibited opposite changes at the mRNA level in the two brain tissues. No KEGG pathways were found with which these DEGs were significantly associated. However, molecular function analysis in the Gene Ontology database showed that genes whose expression was decreased in the striatum but increased in the FC had cytokine, growth factor, and NADPH binding activity (*p* < 0.05), whereas genes whose expression was increased in the striatum but decreased in the FC predominantly had calcium channel activity associations.

### 2.6. Gene Regulatory Networks Characterizing Common and Specific Effects of Ischemia on Striatum and FC Cells 24 h After tMCAO

We selected the genes that participated in the maximum number of pathways, both common (PC1) and unique to each brain region (PC2 and PC3). The *Mapk1* gene was involved in the maximum number (76) of pathways from PC1, overlapping between the FC and striatum tissues. The gene had decreased expression in both the striatum and FC. The second place was taken by the *Prkacb* gene, participating in the regulation of 48 pathways. It also had decreased expression in both tissues. In total, 418 DEGs were involved in two or more PC1 pathways. For clarity of network visualization, the first 18 genes involved in the regulation of more than 30 pathways each were selected (Figure 5a). These included genes encoding transcription factors (*Fos*, *Jun*, *Nfkb1*), genes of neurosignaling proteins (*Plcb1*, *Adcy4*, *Adcy7*, *Adcy9*, *Gnai3*), apoptosis (*Tp53*), etc. These genes that showed co-directional changes in expression in both the striatum and FC were nodal and characterized the overall impact of ischemia in both tissues 24 h after tMCAO.

Furthermore, the genes of PC2 were examined. Namely, the *Pik3cd*, *Nras*, *Hras*, and *Pik3r2* genes were involved in the maximum number (15) of pathways from PC2 specific to the striatum. All these genes changed expression only in the striatum. So, *Nras* was up- whereas *Pik3cd*, *Hras*, and *Pik3r2* were downregulated DEGs. In total, 107 DEGs were involved in two or more PC2 pathways. The first 14 genes involved in the regulation of more than five pathways each were selected for the network (Figure 5b). These predominantly included genes encoding proteins of apoptosis (*Bcl2*, *Casp9*), calcium transport (*Prkcb*, *Itpr1*, *Itpr3*), and molecular functions. These genes characterized the specific impact of ischemia in the striatum 24 h after tMCAO. Since they were involved in the maximum number of interactions in the pathways, they could persist as stabilizers of processes in this area of the brain.

Finally, the genes of PC3 were examined. Genes that were unique to the FC were predominantly not hubs of several signaling pathways but only of one pathway in PC3. At the same time, we found 13 genes (*Mapk3*, *Aldh3a2*, *Gss*, *Dgke*, *Plcd1*, *Inpp5a*, *Pip5k1b*, *Pip4k2c*, *Pip4k2b*, *Kras*, *Pik3r1*, *Grin2a*, and *Calm2*) predominantly associated with the lipid metabolism molecular function. These genes were involved in the regulation of more than one pathway each. Among them, only the *Mapk3* gene was involved in three pathways, and the others were involved in two pathways. The genes were selected for the network shown in Figure 5c. These genes were DEGs only in the FC and not in the striatum. They could characterize the specific impact of ischemia in the FC 24 h after tMCAO.

Then, we analyzed the involvement of the DEGs that exhibited opposite changes at the mRNA level in the pathways of the two brain tissues. Of the 54 such genes, we found 24 genes that were involved in the regulation of the pathways from PC1–PC3. For illustration purposes, we selected only genes that were involved in the top of the most significant pathways according to the *Padj* level. As a result, 11 genes (*Mecom*, *Cacna1d*, *Tgfb2*, *Kitlg*, *Cnr1*, *Htr1d*, *Bmp2*, *Cav1*, *Trpc4*, *Kcnj12*, and *Dio2*) were involved in the regulation of one of more pathways and were selected for the network (Figure 6). So, the *Cacna1d*, *Kcnj12 Tgfb2*, *Cav1*, *Mecom*, *Kitlg*, and *Bmp2* genes had associations with any of the five pathways of PC1 (glutamatergic synapse, cholinergic synapse, proteoglycans in cancer, pathways in cancer, and MAPK signaling pathway). Also, the *Cav1*, *Cacna1d*, *Trpc4*, *Tgfb2*, and *Dio2* genes had associations with any of the five pathways of PC2 (pathways of neurodegeneration for multiple diseases, bacterial invasion of epithelial cells, GnRH secretion, FoxO signaling, and thyroid hormone pathway). It should be noted that only two pathways from PC3 (renal cell carcinoma and neuroactive ligand–receptor interaction) were found to have associations with the 54 studied genes. In particular, the *Tgfb2*, *Cnr1*, and *Htr1d* genes had associations with any of the listed pathways. These genes were also included in the network (Figure 6).

Network analysis revealed that *Tgfb2* and *Cacna1d* genes had involvements in maximal number (five) of the most significant pathways from different clusters. So, the *Tgfb2* gene had three involvements in PC1, one involvement in PC2, and one involvement in PC3. Concomitantly, the *Cacna1d* gene had three involvements in PC1 and two involvements in PC2. Both these genes were downregulated in the striatum but upregulated in the FC 24 h after tMCAO. There were genes that participated only in the pathways of one cluster: PC1 (*Mecom*, *Kitlg*, *Bmp2*, *Kcnj12*), PC2 (*Htr1d*, *Dio2*), and PC3 (*Cnr1*, *Trpc4*). The network shows the spectrum of genes and functional-related systems whose activities may change in brain tissues with different damage degrees 24 h after tMCAO.

## 3. Discussion

In this study, we analyzed the transcriptomes of the striatum 24 h after tMCAO. Tissue predominantly contained the infarct area and partly contained the penumbra area, according to MRI and HE data. More than 4000 DEGs were identified by us in the striatum. Previously, we showed that IR predominantly alters the expression of more than 3000 genes in SO animals in the frontal cortex (FC) containing penumbra and viable cells [31]. Therefore, here, we additionally compared the effect of IR on the transcriptomes of rat brain regions with different degrees of ischemic damage 24 h after tMCAO. We revealed that DEG expression profiles 24 h after tMCAO overlapped between tissues in 2609 genes. We assigned pathways that were associated with such DEGs as PC1. Most of the overlapped DEGs showed a co-directional expression profile in brain regions with different degrees of damage. They were associated with glutamatergic synapses (e.g., *Gng12*, *Gnai3*, *Gng5*, *Grik5*), lipids and atherosclerosis (e.g., *Cd14*, *Ldlr*, *Plcb1*, *Vcam1*), and MAPK (e.g., *Map3k6*, *Gadd45g*, *Ngfr*, *Jun*) and other signaling pathways. Furthermore, the most significant increase in gene expression in both tissues was observed in the group of pathways associated with the immune system (e.g., *Cd86*, *RT1-Da*, *Tgfb1*, *Cxcr4*), as well as ribosome biogenesis (e.g., *Eif6*, *Rpp38*, *Utp15*, *Mphosph10*), apoptosis (e.g., *Bcl2a1*, *Casp3*, *Tp53*, *Bak1*), VEGF (e.g., *Ptgs2*, *Pla2g4a*, *Nos3*, *Sphk1*), and cellular senescence (e.g., *Chek1*, *Rras*, *Gadd45b*, *Nfkb1*). Concomitantly, the most significant decrease in gene expression was in the group of pathways associated with neurotransmission (e.g., *Gabrd*, *Drd1*, *Slc17a7*, *Grik5*, *Grin3a*, *Plcb1*, *Adcy5*, *Htr2c*, *Kcnq3*). The involvement of these systems is consistent with events of the ischemic cascade, including events of glutamate toxicity [12,13], oxidative stress, and mitochondrial dysfunction [14,15], as well as inflammatory reactivity [16]. For example, *Mapk1*, *Prkacb*, *Fos*, *Jun*, *Nfkb1*, *Tp53*, and *Adcy4* were involved in the maximum number of pathways from PC1 overlapping between the FC and striatum tissues. The significance of most of these genes is well known after cerebral ischemia [42,43,44]. Additionally, the involvement of genes in ribosome biogenesis and apoptosis in the IR response was previously revealed by us in subcortical structures [21].

Interestingly, more than one thousand DEGs were specific for each studied tissue. Namely, 1800 and 1165 genes were individually found as DEGs for the striatum and FC, respectively. Consequently, PC2 and PC3 presented unique gene expression effects due to IR in every brain region. The spectrum of genes and associated pathways for the striatum (PC2) were more numerous. Recent data from Rowe et al. reported that the expression profile of the striatum was the most distinct and was isolated from all other brain regions across every injury pattern and timepoint following traumatic injury [45]. In our data, the gene functional annotations of PC2 predominantly included calcium, neurotrophin, MAPK, thyroid and parathyroid hormone, GABAergic, dopamine, choline, and other pathways that were associated with downregulated DEGs in the striatum. Additionally, there were proteasome, cellular senescence, natural killer cell-mediated cytotoxicity, platelet activation, and other pathways that were predominantly associated with upregulated DEGs in the striatum. Additionally, the PC2 pathways unique to striatal DEGs included GnRH secretion, cell cycles, antigen processing and presentation, DNA replication, phagosomes, and others. It should be noted that we identified the involvement of some of them (antigen processing and presentation, phagosomes, thyroid hormone synthesis, DNA replication) in the subcortical structures of the rat brain 24 h after tMCAO [21]. It is well known that thyroid hormones are important for the regulation of the development and differentiation of neurons and neuroglia [46]. In addition, the relationship between thyroid hormones and stroke outcomes is actively discussed [47,48].

We selected genes that were involved in the maximum number of pathways from PC2 specific for the striatum. These predominantly included genes encoding proteins of immune (e.g., *Pik3cd*), apoptosis (e.g., *Bcl2*, *Casp9*), and calcium transport (e.g., *Prkcb*, *Itpr1*, *Itpr3*) molecular functions. They could characterize the specific impact of ischemia in striatum 24 h after tMCAO. It should be noted that some of these genes were also associated with thyroid hormone synthesis. Furthermore, it has recently been shown that the inhibition of PI3Kδ which is encoded by the *Pik3cd* gene alleviates brain injury during cerebral ischemia–reperfusion via suppressing pericyte contraction in a TNF-α-dependent manner [49]. *Pik3cd* was found in a hub of 15 pathways in the striatum, so the gene could persist as a stabilizer of processes in this area of the brain under IR conditions.

PC3 reflected a specific transcriptome response in the FC. PC3 predominantly included neuroactive ligand–receptor interaction and cysteine and methionine metabolism with downregulated DEGs (e.g., *Grin2a*, *Htr1a*, *Adra1b*, *Npy5r*, *Gabra1*, *Adi1*, *Mat2b*, *Bhmt*) in IR-f vs. SO-f. Furthermore, PC3 pathways including the biosynthesis of cofactors, pentose and glucuronate interconversions, IL-17, and ferroptosis were predominantly associated with upregulated DEGs (e.g., *Gss*, *Il1b*, *Ccl11*, *Acsl1*, *Xylb*, *Akr1b8*) in IR-f vs. SO-f. It should be noted that *Mapk3* had involvements in the maximal number of pathways from PC3. Mitogen-activated protein kinase 3, encoded by the *Mapk3* gene, is involved in the regulation of apoptosis during IR in various tissues [50,51,52]. A number of studies have identified *Mapk3* as a neuroprotective target using network pharmacological analysis [53,54,55]. It is possible that the specificity of the gene expression profile for the FC may determine the pathways for correcting the state of FC cells after stroke in the penumbra, including through the regulation of apoptosis.

Recently, the spatial and single-cell transcriptome profiling of male mouse brains during the first week of injury was performed by Zucha et al. Cortical gene expression was severely impaired, as determined by inflammation and cell death in the nucleus lesions, as well as the formation of a glial scar organized by several cell types at the periphery [32]. The results were obtained 1, 3, and 7 days after pMCAO model conditions. The model simulated clinical cases of stroke that do not receive timely treatment, resulting in the blocked artery remaining impassable [56]. The results provided valuable insight into spatial processes in the brain after stroke, but the process of thrombus recanalization and reperfusion was underestimated. We made a comparison between our RNA-Seq data (DEGs overlapped between striatum and FC, as well as specific for each tissue) and results Zucha et al. for each gene set. Hundreds of genes in our study were found in the results of Zucha et al. Of these, 362 striatum-specific genes (e.g., *Nefl*, *Rgs4*, *Rasgrp1*, *Mapk10*, *Sv2a*, *Slc17a6*, *Cacna2d2*, *Snap25*, *Cct8*) were found for any caudoputamen, hippocampus, thalamus, and hypothalamus. In addition, FC-specific genes included genes overlapping with 215 isocortex and cortical subplate genes, including *Chn1*, *Vsnl1*, *Mat2b*, *Calm2*, *Cort*, *Gria2*, *Pld3*, and *Numbl*. The genes that we identified as overlapping between our and Zucha et al.’s data are shown in Appendix A. However, more than 2000 genes (e.g., *Cln6*, *Chrna2*, *Ntrk1*, *Npas1*, *Chrna5*, *Npy5r Timp1*, *H19*, *Ptx3*, *Bpifb1*, *Serpine1*) were found by us as unique DEGs in our study. Their reveal may be a result of the features of single-cell or full-tissue transcriptome methods, as well as bioinformatics. Concomitantly, the genes identified may be related to the specificity of the rat model, as a larger animal than the mouse, and also reflect the effect of reperfusion, which was taken into account by our tMCAO model of reversible ischemia but not the pMCAO of Zucha et al. Thus, our results are partially consistent with previous data but significantly complement the existing landscape of spatial brain transcriptomics following experimental ischemic stroke.

Interestingly, 54 DEGs (*Tgfb2*, *Cacna1d*, *Cnr1, Dio2*, and others) exhibited opposite changes at the mRNA level in the striatum and FC 24 h after tMCAO. Many of them were included in PC1–PC3; therefore, the activity of these genes could control both general and specific metabolic processes for the striatum and FC during IR. Network analysis revealed that *Tgfb2* and *Cacna1d* had involvements in the maximal number of pathways from different clusters. Both of these genes were downregulated in the striatum but upregulated in the FC 24 h after tMCAO. In particular, *Tgfb2* encoded transforming growth factor-beta 2 with many cellular functions including T-cell activity control. Recently, Rowe et al. established the *Tgfb2* gene as a DEG in the striatum and thalamus following traumatic injury [45]. Also, the expression of the *Cnr1* genes of the endocannabinoid system was modulated in hypoxia–ischemia-induced mice by octadecylpropyl sulfamide [57] and under tMCAO conditions [58]. The *Dio2* gene, encoding iodothyronine deiodinase 2 (D2), was also among the genes that divergently changed expression in the FC and striatum. Enzymes are critically important in triiodothyronine (T3) homeostasis [59]. The thyroid hormone signaling pathway was specific to the striatum response, in accordance with our results 24 h after tMCAO. The hypothalamic–pituitary–thyroid axis has been shown to be a critical regulator of stroke recovery. Moreover, allelic variants of thyroid axis genes can predict the emotional consequences of stroke [60]. A review of the literature shows the involvement of the listed genes in the processes of the regulation of damage and restoration of neurological functions. Thus, these genes can reflect changes in brain tissues with different damage degrees 24 h after tMCAO. They can be hubs in switching the cell program during the development of ischemia and compensatory mechanisms of regeneration in brain regions after stroke.

The obtained results can be important for studying stroke in humans. Many studies report about exploring biomarkers for ischemic stroke through integrated gene expression analysis [61,62,63,64]. Previously, we analyzed genes that alter expression in tMCAO conditions and found associations of their human orthologs with the risk and outcome of ischemic stroke. In particular, single-nucleotide polymorphisms (SNPs) in *LGALS3* (rs66782529), *LSP1* (rs907611), *GPNMB* (rs858239), and *TAGLN* (rs494356) were associated with various parameters of functional outcomes of ischemic stroke in the Russian population, whereas SNPs in *PTX3* (rs62278647 and rs2316710) and *PDPN* (rs1261025) were associated with the disease risk [65,66]. Thus, it was shown how genes from a rat model of cerebral ischemia can be useful for associating certain genetic variations with ischemic stroke in humans. Furthermore, the identification of novel biomarkers for ischemic stroke through integrated bioinformatics analysis and machine learning contributes to deepening our knowledge of ischemic stroke biomarkers and provides reference data for the prognosis and diagnosis of ischemic stroke [67]. Recently, Zhang et al. revealed the potential role of hub metabolism-related genes (*FBL*, *HEATR1*, *HSPA8*, *MTMR4*, *NDUFC1*, *NDUFS8*, and *SNU13*) and their correlation with immune cells in acute ischemic stroke [68]. It is possible that the genes identified in our study, in accordance with their expression profile in the striatum and FC, will be useful in clinical management and diagnostics, including their role as targets for new effective anti-stroke reagents.

Summarily, DEGs in the infarct-associated striatum and penumbra-associated FC were analyzed in our study. On the one hand, the tissues were affected by ischemia, but on the other hand, they differed in the degree of damage and cellular composition. As a result, the transcriptomic response of each tissue had both common and specific features under IR. Furthermore, the spectrum of genes and functional-related systems whose activities may have changed in brain tissues with different damage degrees 24 h after tMCAO were revealed. In particular, DEGs that exhibited opposite changes at the mRNA level in the two brain tissues after tMCAO were revealed. There are a number of limitations on our study. It should be noted that all results were obtained using young male rats and, therefore, age- and sex-related bias could exist. Furthermore, studies of other timepoints can provide molecular bases of dynamic changes after stroke in brain. These limitations may be overcome in the future.

## 4. Materials and Methods

### 4.1. Animals

White 2-month-old male rats of the Wistar line (weight, 200–250 g) were obtained from AlCondi, Ltd. (Moscow, Russia), as previously described [69]. The animals were divided into groups: “sham operation” (SO) and “ischemia–reperfusion” (IR). The IR animals were subjected to the transient middle cerebral artery occlusion (tMCAO) model. Each experimental group included at least five animals.

### 4.2. tMCAO Model

The tMCAO model with 90 min occlusion was used in accordance with the method of Koizumi et al. [70] under magnetic resonance imaging (MRI) and histological examination (HE), as described previously [31,71]. The tMCAO and HE details are described in Appendix A, respectively. Temporal isoflurane anesthesia was used for the animals during operation, occlusion, MRI, and decapitation. The rats were decapitated 24 h after the beginning of tMCAO/sham operations.

### 4.3. Sample Collection and RNA Isolation

The ipsilateral fragments of the region of the frontal cortex (FC) +2–+5 mm from bregma, as well as the ipsilateral striatum (Appendix A), were previously obtained [31]. The resulting striatum samples of the SO and IR groups were named SO-s and IR-s, whereas the FC samples were named SO-f and IR-f, respectively. All SO-s, IR-s, SO-f, and IR-f samples were placed in RNAlater (Ambion, Austin, TX, USA) solution for 24 h at 4 °C and stored at −70 °C. Then, total RNA was isolated, and RNA integrity was checked using capillary electrophoresis (Experion, BioRad, Hercules, CA, USA), as previously described [30]. The RNA integrity number (RIN) was at least 9.0.

### 4.4. RNA-Seq

The polyA fraction of the total RNA was obtained by RNA-Seq analysis using an Illumina HiSeq 1500 instrument, as previously described [30]. At least 10 million reads (1/50 nt) were generated. The RNA-Seq analysis was performed with the participation of OOO Genoanalytika, Moscow, Russia.

### 4.5. cDNA Synthesis and Real-Time Reverse Transcription Polymerase Chain Reaction (RT-PCR)

cDNA was synthesized using oligo (dT)_18_ primers, as previously described [30]. The PCR primers were selected using the OLIGO Primer Analysis Software version 6.31 and were synthesized by the Evrogen Joint Stock Company, Moscow, Russia (Appendix A). Each cDNA sample was analyzed three times using RT-PCR, as previously described [30].

### 4.6. RNA-Seq Data Analysis

Three animals (n = 3) were included in each of the comparison groups (SO, IR) for RNA-Seq experiments. Cuffdiff/Cufflinks software (version 2.2.1) was used for gene annotations, as previously described [30]. The levels of mRNA expression were measured using the Cuffdiff program, as previously described [30]. Only the genes that exhibited >1.5-fold changes in expression and had *p*-values (*t*-test) adjusted using the Benjamini–Hochberg procedure lower than 0.05 (*Padj* < 0.05) were considered.

### 4.7. Real-Time RT-PCR Data Analysis

The relative gene expression was calculated using Relative Expression Software Tool (REST) v. 2005 software (gene-quantification, Freising-Weihenstephan, Bavaria, Germany) [72,73], as previously described [30]. The values were calculated as Ef^Ct(ref)^/Ef^Ct(tar)^, where Ef is the PCR efficiency, Ct(tar) is the average threshold cycle (Ct) of the target gene, Ct(ref) is the average Ct of the reference gene, and Ef^Ct(ref)^ is the geometric average Ef^Ct^ of the reference gene. The efficiency values for all PCR reactions were in the range 1.89 to 2.01 (Appendix A). The reference gene *Gapdh* was used to normalize the expression of the cDNA samples. Each comparison group consisted of five animals. Significant differences were considered at *p* < 0.05.

### 4.8. Functional Analysis

The functions of the differentially expressed mRNAs (DEGs) were annotated using the Database for Annotation, Visualization, and Integrated Discovery (DAVID) (2021 Update) [74]. Only functional categories that had *Padj* < 0.05 were considered. Hierarchical cluster analysis of the DEGs was performed using Heatmapper (Wishart Research Group, University of Alberta, Ottawa, ON, Canada) [75]. A volcano plot was constructed using Microsoft Excel (Microsoft Office 2010). Cytoscape 3.9.2 software (Institute for Systems Biology, Seattle, WA, USA) [76] was used to visualize the regulatory network.

### 4.9. Availability of Data and Material

The RNA-sequencing data were deposited in the Sequence Read Archive database under the accession code PRJNA1119923 (SAMN41664920-SAMN41664925, https://dataview.ncbi.nlm.nih.gov/object/PRJNA1119923?reviewer=plp0lftpig94lmq8vmup84at3d, accessed on 25 June 2024) [77], PRJNA1128447 (SAMN42050762-SAMN42050767, https://dataview.ncbi.nlm.nih.gov/object/PRJNA1128447?reviewer=vkci32ofqame6mkpbivjt1pqn8, accessed on 25 June 2024) [78].

## 5. Conclusions

In conclusion, genes that are associated with the action of IR in rat brain regions with different degrees of ischemic damage were revealed. Hundreds of genes both overlapping and specific to the penumbra-associated FC and infarct-associated striatum 24 h after tMCAO were identified. Functional analysis revealed genome-wide associations of DEGs with inflammatory, neurosignaling, and metabolic systems in both studied tissues. Concomitantly, unique DEGs identified in striatum were mainly associated with a wide range of neurosignaling pathways, as well as cellular senescence, antigen processing and presentation, DNA replication, and thyroid and parathyroid hormone systems, whereas unique DEGs identified in the FC were predominantly related to the biosynthesis of cofactors ferroptosis and apoptosis, as well as IL-17 responses. Furthermore, we revealed DEGs that exhibited opposite changes at the mRNA level in the two brain tissues after tMCAO. It is possible that the switching of cellular programs in the brain during ischemia was associated with a change in the activity of these genes. Thus, the spatial regulation of the ischemic process in the ipsilateral hemisphere of rat brain at the transcriptome level was revealed after tMCAO in rats. Further studies at the cellular level are needed in the future to determine the role of these genes in the regulation of molecular signatures, signaling pathways, and functional networks in cells under ischemia conditions. Approaches can include switching genes on and off at the RNA and protein levels, structural biology studies, and integrative studies with machine leaning. We believe that the targeted adjustment of the genome responses identified can be the key for the induction of regeneration processes in brain cells after stroke and following its treatment.

## Figures and Tables

**Figure 1 ijms-26-02347-f001:**
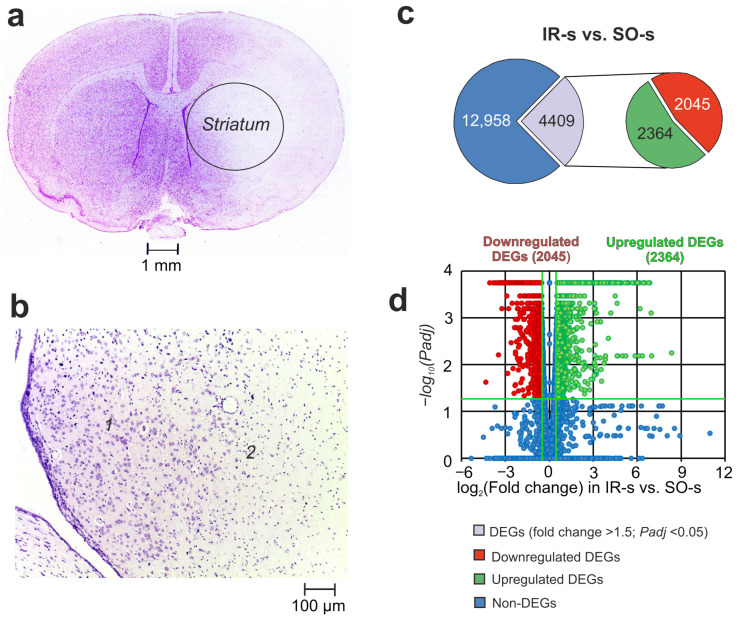
The morphometry zones of Nissl-stained neurons and RNA-Seq analysis of the effect of ischemia on the transcriptome of the striatum of rats 24 h after tMCAO. (**a**) A serial coronal brain section at a level of +1.0 mm from the bregma. (**b**) High-magnification images of the striatum of the right (ipsilateral) hemisphere at a level of +0.48 mm from the bregma. The zone of the penumbra and normal tissues is shown by “1”, whereas the zone of necrotic tissue is shown by “2”. (**c**) RNA-Seq results for IR-s vs. SO-s. The numbers in the diagram sectors indicate the quantity of DEGs. (**d**) Volcano plots show a comparison of the gene distribution between the IR-s and SO-s groups. Upregulated and downregulated DEGs are represented as red and green dots, respectively (fold change > 1.50; *Padj* < 0.05). Non-differentially expressed genes (non-DEGs) are represented as blue dots (fold change ≤ 1.50; *Padj* ≥ 0.05). Each group includes three rats.

**Figure 2 ijms-26-02347-f002:**
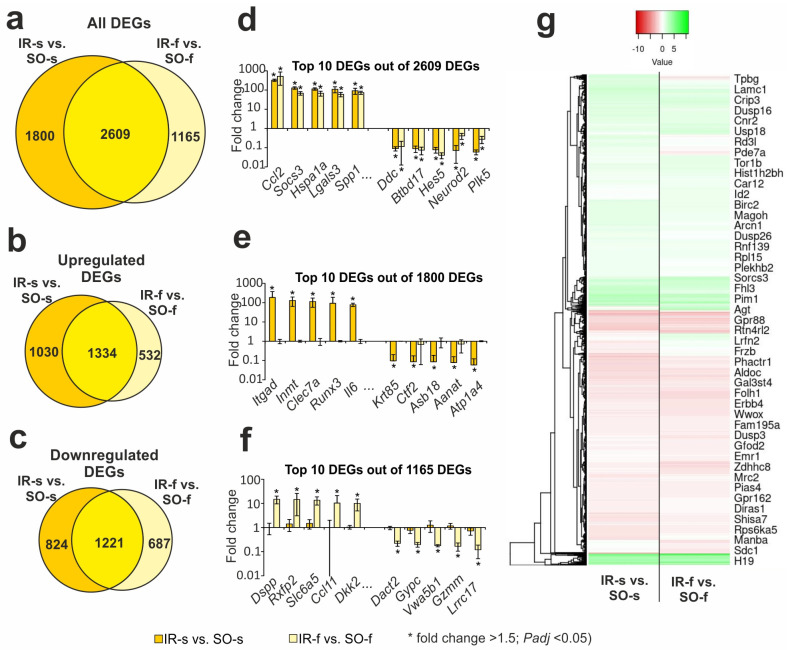
The RNA-Seq analysis of ischemia-induced gene expression changes in the striatum and FC 24 h after tMCAO. (**a**–**c**) Venn diagrams represent results obtained in comparisons between IR-s and SO-s in the striatum and IR-f and SO-f in the FC. All (**a**), upregulated (**b**), and downregulated (**c**) DEGs are shown for comparison. (**d**–**f**) The top ten genes among the 2609 overlapped DEGs (**d**), 1800 DEGs specific for IR-s vs. SO-s (**e**), and 1165 DEGs specific for IR-s vs. SO-s (**f**) are shown in the Venn diagram (**a**), respectively. DEGs were chosen with the greatest fold changes in the IR-s vs. SO-s (**d**,**e**) and IR-f vs. SO-f (**f**) comparison groups. (**g**) Hierarchical cluster analysis of all DEGs in IR-s vs. SO-s and IR-f vs. SO-f, where each row represents a DEG; n = 3 per group. Only those genes with a cut-off >1.5 and *Padj* < 0.05 were selected as significant results. The data are presented as the mean ± standard error of the mean (SEM).

**Figure 3 ijms-26-02347-f003:**
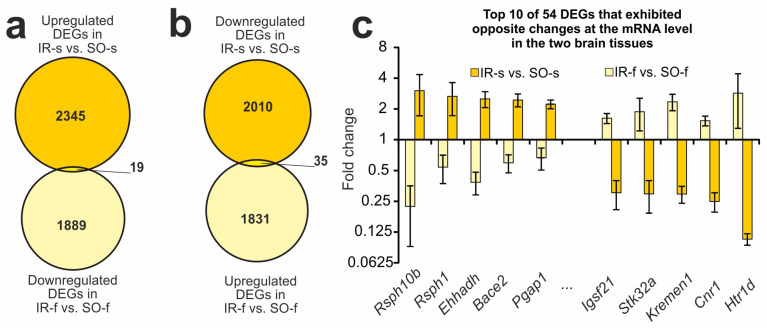
DEGs exhibited opposite changes at the mRNA level in the two brain tissues 24 h after tMCAO. (**a**,**b**) The Venn diagrams represent results obtained in comparisons between the upregulated DEGs in the IR-s and SO-s and downregulated DEGs in the IR-f and SO-f groups (**a**), as well as in comparisons between the downregulated DEGs in the IR-s and SO-s and upregulated DEGs in the IR-f and SO-f groups in the striatum (**b**). The cut-off for gene expression changes was 1.50-fold. Only genes with *Padj* < 0.05 and a cut-off > 1.5 were selected for analysis. (**c**) The top ten overlapped DEGs that changed expression in the opposite direction in the IR-s vs. SO-s and IR-f vs. SO-f pairwise comparisons. Data are presented as the mean ± SEM.

**Figure 4 ijms-26-02347-f004:**
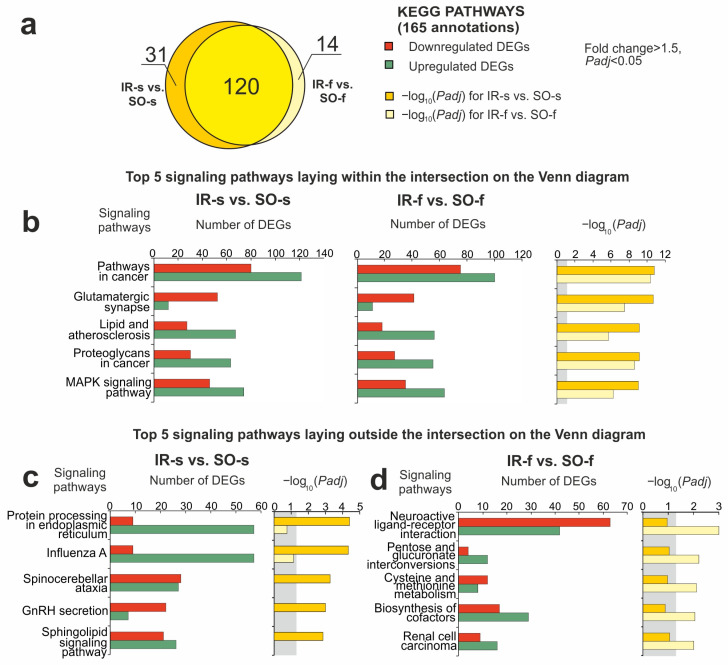
Overlapped and unique signaling KEGG pathways associated with DEGs in the striatum and FC of rats 24 h after tMCAO. The pathway enrichment analysis of DEGs was carried out according to DAVID (2021 Update). (**a**) A schematic comparison of DEG-related annotations in the IR-s vs. SO-s and IR-f vs. SO-f pairwise comparisons using a Venn diagram. The number of annotations is indicated using numbers on the chart segments. (**b**–**d**) The most significant pathways among the 120 overlapped pathways (**b**), among 31 pathways that were unique for IR-s vs. SO-s (**c**), and among 14 pathways that were unique for IR-f vs. SO-f (**d**). The number of upregulated (green) and downregulated (red) DEGs in the two pairwise comparisons—IR-s vs. SO-s (**b**,**c**) and IA-s vs. IR-s (**b**,**d**)—as well as the corresponding *Padj* values are presented. Only DEGs and pathways with *Padj* < 0.05 were selected for analysis, with n = 3 animals per group. *Padj* ≥ 0.05 is enclosed in the gray background.

**Figure 5 ijms-26-02347-f005:**
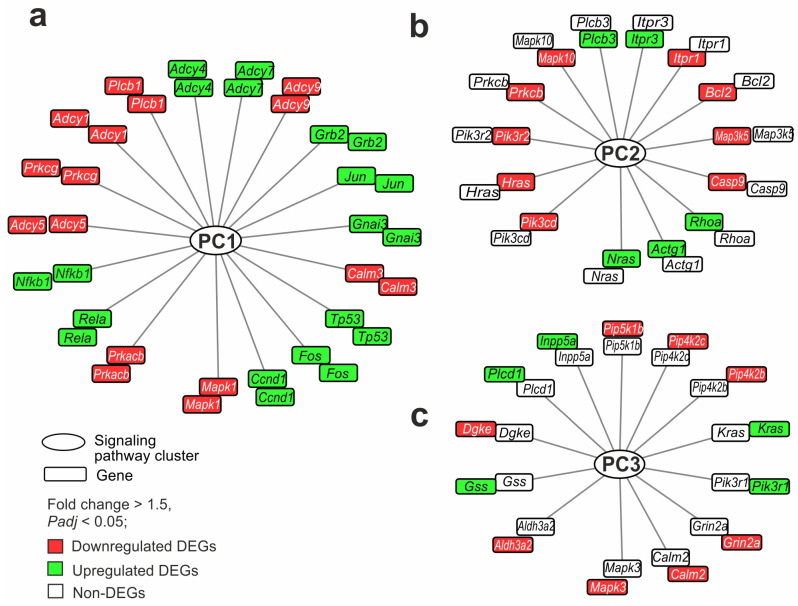
Gene regulatory networks demonstrating common and specific effects of ischemia on striatum and FC cells 24 h after tMCAO. (**a**–**c**) The genes that overlapped between IR-s and SO-s and IR-f and SO-f (**a**), were DEGs in IR-s vs. SO-s but non-DEGs in IR-f vs. SO-f (**b**), and were DEGs in IR-f vs. SO-f but non-DEGs in IR-s vs. SO-s (**c**) are shown. Furthermore, each network includes genes that participate in the maximum number of pathways: common (PC1) (**a**) and unique to the striatum (PC2) (**b**) and FC (PC3) (**c**). The genes in each network are arranged in two rings. Each ring includes the same genes, but the color in the inner ring identifies DEGs in IR-s vs. SO-s and the color in the outer ring determines the DEGs in IR-f vs. SO-f. Lines connecting genes and PCs indicate the participation of the protein products of the genes in the functioning of the pathway of the PCs. DAVID v2021 software was used for annotation of DEG functions based on the KEGG database. The network was constructed with the help of Cytoscape 3.9.2 program.

**Figure 6 ijms-26-02347-f006:**
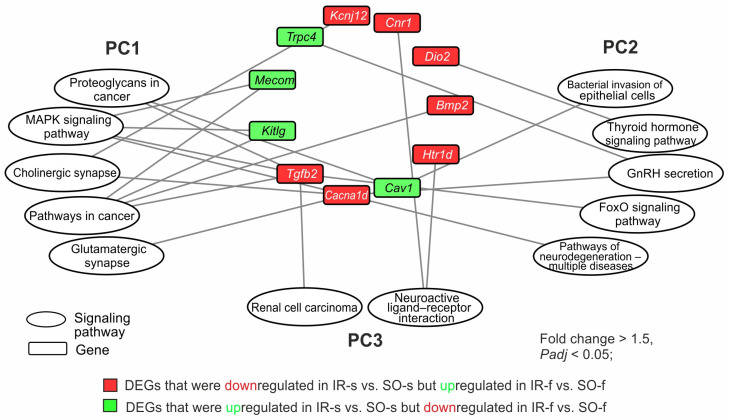
The involvement of genes that changed their mRNA level in the opposite direction in the IR-s vs. SO-s and IR-f vs. SO-f pairwise comparisons in the presentation of the pathways of PC1–PC3 with minimal *Padj*-values. The pathway enrichment analysis of DEGs was carried out according to DAVID (2021 update). Only DEGs and pathways with *Padj* < 0.05 were selected for analysis, with n = 3 animals per group. The network was constructed using Cytoscape 3.9.2 (Institute for Systems Biology, Seattle, WA, USA).

## Data Availability

Publicly available datasets were analyzed in this study. These data can be found here: [77].

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
