# Peer review of "Differentially Expressed Genes in Rat Brain Regions with Different Degrees of Ischemic Damage"

_ijms, 2025, doi:10.3390/ijms26052347_

Round 1
Reviewer 1 Report
Comments and Suggestions for Authors
The manuscript titled “Differential Expressed Genes in Rat Brain Regions with Different Degree of Ischemic Damage” by Filippenkov, I.B.; et al. is a scientific work where the authors addressed the gene identification in different brain regions in rats with ischemia stroke. The most relevant findings gathered in this research could serve to gain crucial knowledge about those underlying molecular mechanisms that leat to this brain disease. The manuscript is generally well-written and this is a topic of growing interest.
However, it exists some points that need to be addressed (please, see them below detailed point-by-point) to improve the scientific quality of the submitted manuscript paper before this article will be consider for its publication in the International Journal of Molecular Sciences.
1) The authors should consider to add “brain damage” in the keyword list.
2) Introduction. “Ischemic stroke is a multifactorial disease that leads to brain tissue damage (…) Ten of millions of people suffer from stroke every year” (lines 39-40). Could the authors provide quantitative data insights according to the disability-adjusted life years (DALYs) concerning ischemic strokes? This will greatly benefit the potential readers to better understand the significance of the work devoted in this research.
3) Then, the authors should also highlight how ischemia can accelerate the onset of some brain malignancies as Alzheimer’s disease. In this context, the deregulation of ionic strengh, protein concentration [1] and certain divalent cations [2] can trigger it.
[1] https://doi.org/10.3390/ijms222212382
[2] https://doi.org/10.3390/biom14091091
4) Results. Figure 1, panel b. The lateral scale bar of the histological image should be also added.
5) Did the authors carry out genome-wide associated studies (GWAS) to associate specific genetic variations with this particular brain disease? Some information needs to be furnished in this regard.
6) Figure 4. Did the authors conduct some statistical analysis to discern if the observed differences among the examined signaling pathways are statistically significant?
7) “5. Conclusions” (lines 501-518). This section perfectly remarks the most relevant outcomes found by the authors in this work and also the promising future prospectives. It may be advisable to add a brief statement to remark the potential future action lines to pursue the topic covered in this research.
Author Response
Response to the comments of Reviewer 1 to Manuscript ID: ijms-3498309
Authors:
We are very grateful to the Reviewer 1 for the review and constructive comments. We carefully considered the comments of the Reviewer 1 and attached the answers to all comments.
Reviewer 1:
- The authors should consider to add “brain damage” in the keyword list.
Authors:
In accordance with the Reviewer’s recommendation, changes were added in the text of the Manuscript (line 34 in Mark-up Copy_R1).
Reviewer 1:
- “Ischemic stroke is a multifactorial disease that leads to brain tissue damage (…) Ten of millions of people suffer from stroke every year” (lines 39-40). Could the authors provide quantitative data insights according to the disability-adjusted life years (DALYs) concerning ischemic strokes? This will greatly benefit the potential readers to better understand the significance of the work devoted in this research.
Authors:
In accordance with the Reviewer’s recommendation, changes were added in the text of the Manuscript (line 40-44 in Mark-up Copy_R1).
Reviewer 1:
- Then, the authors should also highlight how ischemia can accelerate the onset of some brain malignancies as Alzheimer’s disease. In this context, the deregulation of ionic strengh, protein concentration [1] and certain divalent cations [2] can trigger it.
[1] https://doi.org/10.3390/ijms222212382
[2] https://doi.org/10.3390/biom14091091.
Authors:
In accordance with the Reviewer’s recommendation, changes were added in the text of the Manuscript (lines 44-49 in Mark-up Copy_R1).
Reviewer 1:
- Figure 1, panel b. The lateral scale bar of the histological image should be also added.
Authors:
In accordance with the Reviewer’s recommendation, changes were added in the Figure 1.
Reviewer 1:
- Did the authors carry out genome-wide associated studies (GWAS) to associate specific genetic variations with this particular brain disease? Some information needs to be furnished in this regard.
Authors:
Previously, we analyzed genes that alter expression in tMCAO conditions and found associations of their human orthologues with the risk and outcome of ischemic stroke. In particular, single nucleotide polymorphisms (SNPs) in LGALS3 (rs66782529), LSP1 (rs907611), GPNMB (rs858239), and TAGLN (rs494356) were associated with various parameters of functional outcomes of ischemic stroke in the Russian population, whereas SNPs in PTX3 (rs62278647 and rs2316710) and PDPN (rs1261025) were associated with the disease risk [65,66]. Thus, it was shown how genes from a rat model of cerebral ischemia can be useful for associating certain genetic variations with ischemic stroke in humans.
In accordance with the Reviewer’s recommendation, changes were added in the text of the Manuscript (lines 454-462 in Mark-up Copy_R1).
Reviewer 1:
- Figure 4. Did the authors conduct some statistical analysis to discern if the observed differences among the examined signaling pathways are statistically significant?
Authors:
Only pathways with Padj<0.05 according to the David v.2021 program were selected as significant. The common pathways (Figure 4b) were significant (Padj<0.05) in both IR-s vs. SO-s and IR-f vs. SO-f pairwise comparisons. Figure 4c shows the pathways that were significant for IR-s vs. SO-s (Padj<0.05) but not for IR-f vs. SO-f (Padj ≥ 0.05). Also, Figure 4d shows the pathways that were significant for IR-f vs. SO-f (Padj < 0.05) but not for IR-s vs. SO-s (Padj ≥ 0.05). Padj ≥ 0.05 are enclosed in the gray background.
In accordance with the Reviewer’s recommendation, changes were added in the Figure 4 and text of the Manuscript (line 252 in Mark-up Copy_R1).
Reviewer 1:
- “5. Conclusions” (lines 501-518). This section perfectly remarks the most relevant outcomes found by the authors in this work and also the promising future prospectives. It may be advisable to add a brief statement to remark the potential future action lines to pursue the topic covered in this research.
Authors:
We thank Reviewer 1 for his comment and high appreciation. In accordance with the Reviewer’s recommendation, changes were added in the text of the Manuscript (lines 564-568 in Mark-up Copy_R1).

Reviewer 2 Report
Comments and Suggestions for Authors
According to the authors, genes associated with IR's action in rat brain regions with different degrees of ischemic damage were revealed. Hundreds of genes both overlapped and specific to penumbra–associated FC and infarct-associated striatum at 24h after tMCAO were identified. Functional analysis revealed genome-wide associations of DEGs with inflammatory, neuro-signaling, and metabolic systems in both studied tissues. Concomitantly, unique DEGs identified in the striatum were mainly associated with a wide range of neurosignaling pathways, cellular senescence, antigen processing and presentation, DNA replication, and thyroid and parathyroid hormone systems. In contrast, unique DEGs identified in FC were predominantly related to the biosynthesis of cofactors ferroptosis, apoptosis, and IL-17 responses.
Furthermore, authors reveal DEGs that exhibited opposite changes at the mRNA level in the two brain tissues after revealing tMCAO. Switching cellular programs in the brain during ischemia is likely associated with a change in the activity of these genes. Thus, spatial regulation of the ischemic process in the ipsilateral hemisphere of rat brain at the transcriptome level was revealed after tMCAO in rats.
The manuscript's strength is its modern presentation of research results from the field of molecular biology and its new molecular insights into the molecular pathophysiology of Ischemic Damage.
Weakness of the manuscript: The authors use terms familiar to molecular biologists, and medical professionals are unfamiliar with these terminologies.
Suggested minor corrections:
- In the introductory part of the manuscript, explain the RNA-Seq method in more detail.
- Explain the term differentially expressed genes (DEGs) in the introductory part of the manuscript.
- Explain the pMCAO and tMCAO models in more detail, whether there are any other models and the advantages of the pMCAO and tMCAO models.
- Schematically show the metabolic and signaling and (or) physiological pathways where gene activity changes occur.
- At the end of the manuscript, give the meaning of the abbreviations from the text.
Author Response
Response to the comments of Reviewer 2 to Manuscript ID: ijms-3498309
Authors:
We are very grateful to the Reviewer 2 for the review and constructive comments. We carefully considered the comments of the Reviewer 2 and attached the answers to all comments.
Reviewer 2:
- In the introductory part of the manuscript, explain the RNA-Seq method in more detail.
Authors:
High-throughput RNA sequencing (RNA-Seq) method provides measurement of levels of transcripts and their isoforms in biological samples on a genome-wide scale [38]. Thus, RNA-Seq makes it possible to identify genes that show significant differences in expression levels between two or more groups [39,40].
In accordance with the Reviewer’s recommendation, changes were added in the text of the Manuscript (lines 78-84 in Mark-up Copy_R1).
Reviewer 2:
- Explain the term differentially expressed genes (DEGs) in the introductory part of the manuscript.
Authors:
RNA-Seq makes it possible to identify genes that show significant differences in expression levels between two or more groups [39,40]. These genes are called differentially expressed genes (DEGs).
In accordance with the Reviewer’s recommendation, changes were added in the text of the Manuscript (lines 83-85 in Mark-up Copy_R1).
Reviewer 2:
- Explain the pMCAO and tMCAO models in more detail, whether there are any other models and the advantages of the pMCAO and tMCAO models.
Authors:
Permanent middle cerebral artery occlusion (pMCAO) and transient middle cerebral artery occlusion (tMCAO) are more often used for stroke study[19–24]. The pMCAO model simulates most clinical stroke cases that are not treated promptly, leaving the blocked artery obstructed. At the same time, an important component of tMCAO is reperfusion, developing against the background of blood flow restoration. This model mainly reflects events after anti-stroke treatment with thrombolytic drugs [25].
In accordance with the Reviewer’s recommendation, changes were added in the text of the Manuscript (line 59-67 in Mark-up Copy_R1).
Reviewer 2:
- Schematically show the metabolic and signaling and (or) physiological pathways where gene activity changes occur.
Authors:
Pathways where gene activity changes occur were show on Figures 4, 5 and 6. Maps of all pathways are available in the KEGG database according to the identifiers listed in Supplementary Table S6.
Reviewer 2:
- At the end of the manuscript, give the meaning of the abbreviations from the text.
Authors:
In accordance with the Instructions for Authors and Manuscript Submission Overview, Acronyms/Abbreviations/Initialisms should be defined the first time they appear in each of three sections: the abstract; the main text; the first figure or table. When defined for the first time, the acronym/abbreviation/initialism should be added in parentheses after the written-out form.
Therefore, we cannot add the list of abbreviations at the end of the manuscript. Please understand us. All abbreviations were deciphered in the text. Concomitantly, as a sign of sincere respect for your comment, we are adding the list of abbreviations in response to the comment below.
Abbreviation list:
Ct – threshold cycle; D2 – iodothyronine deiodinase 2; DAVID – Database for Annotation, Visualization and Integrated Discovery; DEGs – differential expressed genes; Ef – the PCR efficiency; FC – frontal cortex; GnRH – Gonadotropin Releasing Hormone; HE – histological examination; IR – ischemia-reperfusion; KEGG – Kyoto Encyclopedia of Genes and Genomes; MRI – magnetic resonance imaging; non-DEGs – not differentially expressed genes; PC – Pathway Cluster; pMCAO – permanent middle cerebral artery occlusion; REST – Relative Expression Software Tool; RIN – RNA integrity number; RNA-Seq – high-throughput RNA sequencing; RT-PCR – Reverse Transcription Polymerase Chain Reaction; SEM – standard error of the mean; SNPs – single nucleotide polymorphisms; SO – sham operation; T3 – triiodothyronine; tMCAO – transient middle cerebral artery occlusion; VEGF – vascular endothelial growth factor.

Reviewer 3 Report
Comments and Suggestions for Authors
The authors aimed to comprehensively detect quantitative changes in genes in different brain regions (frontal cortex vs. striatum) when rats were subjected to ischemia-reperfusion treatment, and to clarify the differences in changes between the two brain regions, illustrating a hypothesis suggesting a possible involvement of the some gene products (in particular, genes whose quantitative changes in both brain regions show opposite directions) in the pathogenesis under ischemic state. The authors discussed the beneficial availability of the information regarding the gene expression profile which understand the states of the brain ischemic damage. This issue is of interest, and impact of their results is strong. My overall concern with the article describing the current available data regarding beneficial availability of the gene expression profile against some post-ischemic symptoms which are available for evaluation of the ischemic stroke and the damage, offer something substantial that helps advance our understanding of effective management which draws novel class of effective reagents available in clinic.
To strengthen authors’ perspectives, the relevance of the findings of this study to known post-ischemic conditions, and in particular to diseases predicted from the changes in the specific gene groups found, should be described in the Discussion section. Otherwise, this study will end up being merely a comprehensive search for gene changes.
Author Response
Response to the comments of Reviewer 3 to Manuscript ID: ijms-3498309
Authors:
We are very grateful to the Reviewer 3 for the review and constructive comments. We carefully considered the comments of the Reviewer 3 and attached the answers to all comments.
Reviewer 3:
- The authors aimed to comprehensively detect quantitative changes in genes in different brain regions (frontal cortex vs. striatum) when rats were subjected to ischemia-reperfusion treatment, and to clarify the differences in changes between the two brain regions, illustrating a hypothesis suggesting a possible involvement of the some gene products (in particular, genes whose quantitative changes in both brain regions show opposite directions) in the pathogenesis under ischemic state. The authors discussed the beneficial availability of the information regarding the gene expression profile which understand the states of the brain ischemic damage. This issue is of interest, and impact of their results is strong. My overall concern with the article describing the current available data regarding beneficial availability of the gene expression profile against some post-ischemic symptoms which are available for evaluation of the ischemic stroke and the damage, offer something substantial that helps advance our understanding of effective management which draws novel class of effective reagents available in clinic.
To strengthen authors’ perspectives, the relevance of the findings of this study to known post-ischemic conditions, and in particular to diseases predicted from the changes in the specific gene groups found, should be described in the Discussion section. Otherwise, this study will end up being merely a comprehensive search for gene changes.
Authors:
The obtained results can be important for studying stroke in humans. Many studies report about exploring biomarkers for ischemic stroke through integrated gene expression analysis [61–64]. Previously, we analyzed genes that alter expression in tMCAO conditions and found associations of their human orthologues with the risk and outcome of ischemic stroke. In particular, single nucleotide polymorphisms (SNPs) in LGALS3 (rs66782529), LSP1 (rs907611), GPNMB (rs858239), and TAGLN (rs494356) were associated with various parameters of functional outcomes of ischemic stroke in the Russian population, whereas SNPs in PTX3 (rs62278647 and rs2316710) and PDPN (rs1261025) were associated with the disease risk [65,66]. Thus, it was shown how genes from a rat model of cerebral ischemia can be useful for associating certain genetic variations with ischemic stroke in humans. Furthermore, identification of novel biomarkers for ischemic stroke through integrated bioinformatics analysis and machine learning contributes to deepening our knowledge of ischemic stroke biomarkers and provides reference data for the prognosis and diagnosis of ischemic stroke [67]. Recently, Zhang et al. revealed the potential role of hub metabolism-related genes (FBL, HEATR1, HSPA8, MTMR4, NDUFC1, NDUFS8, and SNU13) and their correlation with immune cells in acute ischemic stroke [68]. It is possible that the genes identified in our study, in accordance with their expression profile in the striatum and FC, will be useful in clinical management and diagnostics, including their role as targets for new effective anti-stroke reagents.
In accordance with the Reviewer’s recommendation, changes were added in the text of Discussion section of the Manuscript (lines 451-471 in Mark-up Copy_R1).

Round 2
Reviewer 1 Report
Comments and Suggestions for Authors
The authors did a great deal of effort to cover all the suggestions raised by the Reviewers. For this reason, the scientific manuscript quality was greatly improved. I warmly endorse this research for further publication in the International Journal of Moleculat Sciences.
Reviewer 3 Report
Comments and Suggestions for Authors
The authors have addressed properly all the issues raised by reviewers including me. I have no more comments, and recommend that this manuscript is acceptable for publication in the journal IJMS.